# Origin of the unusually strong and selective binding of vanadium by polyamidoximes in seawater

Alexander S. Ivanov [1], Christina J. Leggett[2,4], Bernard F. Parker [2,3], Zhicheng Zhang[2], John Arnold[2,3], Sheng Dai [1], Carter W. Abney[1], Vyacheslav S. Bryantsev [1] & Linfeng Rao[2]

Amidoxime-functionalized polymeric adsorbents are the current state-of-the-art materials for collecting uranium (U) from seawater. However, marine tests show that vanadium (V) is preferentially extracted over U and many other cations. Herein, we report a complementary and comprehensive investigation integrating ab initio simulations with thermochemical titrations and XAFS spectroscopy to understand the unusually strong and selective binding of V by polyamidoximes. While the open-chain amidoxime functionalities do not bind V, the cyclic imide-dioxime group of the adsorbent forms a peculiar non-oxido $V^{5+}$ complex, exhibiting the highest stability constant value ever observed for the $V^{5+}$ species. XAFS analysis of adsorbents following deployment in environmental seawater confirms V binding solely by the imide-dioxodes. Our fundamental findings offer not only guidance for future optimization of selectivity in amidoxime-based sorbent materials, but may also afford insight to understanding the extensive accumulation of V in some marine organisms.

[1] Oak Ridge National Laboratory, Oak Ridge, TN 37831, USA. [2] Lawrence Berkeley National Laboratory, Berkeley, CA 94720, USA. [3] University of California-Berkeley, Berkeley, CA 94720, USA. [4] Present address: U. S. Nuclear Regulatory Commission, Rockville, MD 20852, USA. Alexander S. Ivanov and Christina J. Leggett contributed equally to this work. Correspondence and requests for materials should be addressed to C.W.A. (email: abneycw@ornl.gov) or to V.S.B. (email: bryantsevv@ornl.gov) or to L.R. (email: lrao@lbl.gov)

The oceans are the earth's greatest reservoir of industrially important minerals. Though present at extremely dilute concentrations, many metals such as lithium, vanadium, nickel, gold, and uranium are orders of magnitude more abundant in seawater than on land (Fig. 1a). These raw resources are essential to industrialized society, finding application in both well-established and emerging technologies such as a new generation of lithium-vanadium phosphate batteries[1], precious group metal catalysts, and the production of nuclear fuel for base-load electricity generation[2]. While growing global populations and improved standards of living drive the unchecked consumption of terrestrial mineral resources, the oceans remain an effectively untapped reserve. A thriving industry currently recovers the four most concentrated metal ions in seawater ($Na^+$, $K^+$, $Mg^{2+}$, and $Ca^{2+}$), yet the commercial recovery of less abundant metals, which requires the use of highly selective adsorbents for metal accumulation from trace concentrations, thus far remains elusive. A successful understanding of the origins of metal ion selectivity coupled with engineered design of adsorption sites affording the targeted removal of the desired species could enable access to the unparalleled reserves of dissolved metals in seawater, while simultaneously providing a less energy-intensive and more environmentally friendly approach to metal mining compared to land-based operations[3].

Of all metals dissolved at appreciable concentrations in seawater, the recovery of uranium (U) has received focused attention over the past 60 years. Despite its very low concentration (3.3 p.p.b.), the oceans contain roughly 4 billion tons of uranium and exceed terrestrial ores by nearly three orders of magnitude, affording a near inexhaustible supply of uranium for nuclear power production. While several extraction systems have been tested for the recovery of uranium from seawater, such as hydrous titanium oxide[4], engineered proteins[5], and functionalized polymeric adsorbents[6], cost analysis has revealed that none is economically competitive with terrestrial mining. Although the current state-of-the-art polyamidoxime adsorbents achieve capacities of up to 3.8 g U per kg adsorbent after 56 days of contact with seawater[7,8], many of the adsorbent's properties, including selectivity and recycle, are inadequate for an economically viable recovery system.

The major technical impediment to the development of polyamidoxime adsorbents is the remarkable affinity for pentavalent vanadium ions, V(V), which also exist in seawater at low concentrations (1.9 p.p.b.) and vastly outcompete uranium for adsorption sites[9]. As displayed in Fig. 1b, marine tests indicate

polyamidoxime is much more selective toward vanadium than uranium. Thus, despite recent advances affording improved adsorption capacity, such as ligand grafting variations, systematic tuning of polymer morphology[10] and adsorbent pre-treatment parameters[11], many questions remain regarding the dominant factors that determine the selective recovery of metal ions from seawater. To this end, we have applied complementary techniques spanning the theoretical and experimental small molecule and bulk material regimes in an effort to understand the vanadium selectivity displayed by polyamidoxime adsorbents.

Quantum chemical simulations corroborated by potentiometric titrations and calorimetry reveal that the cyclic imide-dioxime ($H_3IDO$) functional group of polyamidoxime (Fig. 1c) achieves the highest stability constant values ever reported for V (V), exemplified by the formation of a rare non-oxido $V^{5+}$ complex. In contrast, complementary investigations of the open-chain acetamidoxime (HAO) functionality (Fig. 1c) indicate virtually no binding with V(V) over a wide pH range. X-ray absorption fine structure (XAFS) investigations of adsorbents following deployment in natural seawater confirm the proposed V(V) coordination solely by the imide-dioxides even in the presence of a large excess of competing ions, thus validating the extension of small molecule and theoretical results to bulk samples under environmental conditions. It is also worth noting that synthetically produced polyamidoxime materials, with adsorption capacities of ~15 g V per kg adsorbent (Fig. 1b), outperform some marine invertebrates, tunicates, in their ability to harvest vanadium. Tunicates are known to concentrate vanadium to a level more than a million times higher than in the surrounding seawater, owing to the presence of vanabin proteins (Fig. 1d) in their blood cells[12]. Vanadium ions tend to attach to a particular region of vanabin, where they are coordinated by the amine nitrogens of lysine and arginine residues[12]. Despite more than 100 years of intensive research, there is no conclusive theory of how or why these organisms collect so much vanadium, and it remains one of the unsolved mysteries in biology. Therefore, our studies offer not only guidance for future optimization of selectivity in amidoxime-based sorbent materials but could be of importance in understanding the accumulation of high levels of vanadium in some marine organisms.

## Results

**Adsorbent composition.** A typical poly(acrylamidoxime) adsorbent material is formed by a polyolefin trunk polymer with graft

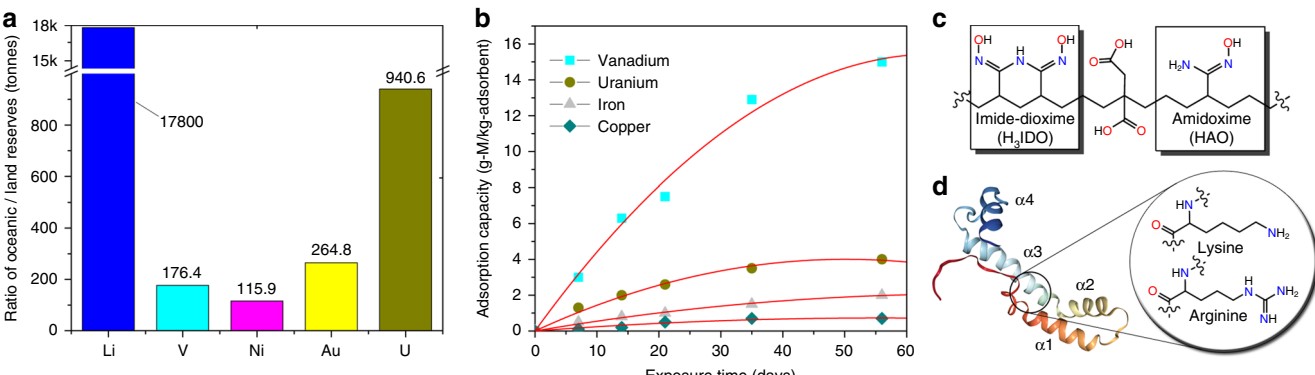

**Fig. 1** Critical metals in seawater and extraction systems. **a** Estimated ratio of the amounts of selected critical metals in the oceans to the terrestrial reserves[49]. **b** Adsorption kinetics of vanadium, uranium, iron, and copper by typical polyamidoxime adsorbents after 56 days of contact with seawater (Sequim Bay, USA) in flow-through columns[7]. **c** Schematic depiction of a small subsection of the polyamidoxime fiber. **d** Cartoon representation of vanabin2 with the highlighted amino acid residues participating in vanadium binding[12,50]

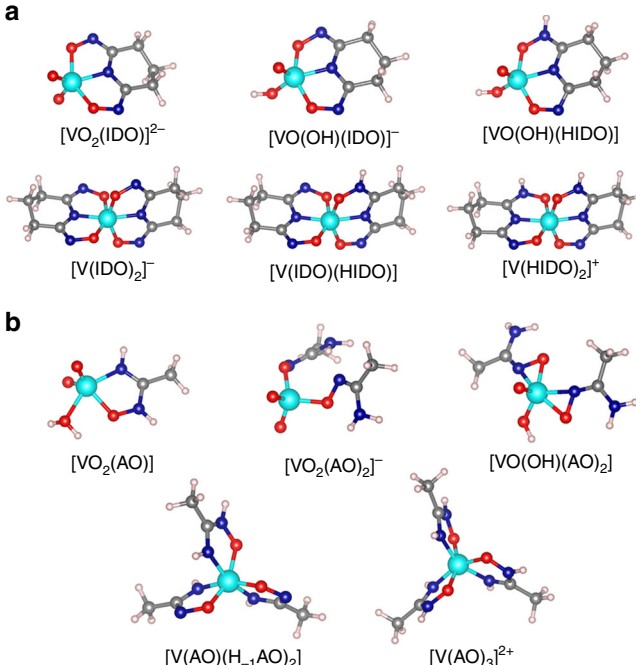

**a**

[VO$_2$(IDO)]$^{2-}$    [VO(OH)(IDO)]$^-$    [VO(OH)(HIDO)]

[V(IDO)$_2$]$^-$    [V(IDO)(HIDO)]    [V(HIDO)$_2$]$^+$

**b**

[VO$_2$(AO)]    [VO$_2$(AO)$_2$]$^-$    [VO(OH)(AO)$_2$]

[V(AO)(H$_{-1}$AO)$_2$]    [V(AO)$_3$]$^{2+}$

**Fig. 2** DFT-optimized geometries of V(V) complexes. **a** V(V) complexes with glutaroimide-dioxime (H$_3$IDO). **b** V(V) complexes with acetamidoxime (HAO). Color legend: V(V), cyan; O, red; N, navy blue; C, gray; H, white

chains composed of amidoximated polyacrylonitrile copolymerized with hydrophilic groups (e.g. acrylic, methacrylic, itaconic or vinyl phosphonic acids). Previous experimental studies indicate that conversion of polyacrylonitrile to polyamidoxime simultaneously generates cyclic imide-dioxime and open-chain amidoxime functionalities (Fig. 1c)[13] and the relative yields of these two functional groups depend strongly on the synthesis conditions[14]. Since the aliphatic carboxylic acid comonomers are weak complexants for vanadium ions[15] and primarily increase fiber hydrophilicity[16], our investigations focused exclusively on binding by the two representative small molecule analogs: glutaroimide-dioxime (H$_3$IDO) and acetamidoxime (HAO).

**Ab initio simulations**. First-principles computations based on density functional theory (DFT) at the M06/SSC/6-311 + + G** level and using the SMD implicit solvation model were performed to elucidate the coordination modes and optimal geometric parameters of the V(V) complexes with H$_3$IDO and HAO. The most stable structures of the complexes in various protonation states in aqueous solution are depicted in Fig. 2, from which key thermodynamic quantities were obtained. The individual stability constant values (log $\beta$) were then theoretically estimated using our recently developed DFT-based protocol[17] augmented by high-level correlated wave function theory calculations (see Methods; Supplementary Note 1), which achieves high accuracy in predicting aqueous log $\beta$ values for V(V) species (root-mean-square error < 0.85 log units). This approach, in principle, allows us to deduce the most prevalent species in solution by relying solely on the ab initio results, as experimental parameters are not required. The theoretically determined log $\beta^{\text{theor}}$ values for the formation of vanadate species with H$_3$IDO and HAO are summarized in Table 1.

According to the generated species distribution diagram (Supplementary Fig. 1), constructed by incorporating log $\beta^{\text{theor}}$

of the V(V)/H$_3$IDO complexes from Table 1 along with the experimental hydrolysis constants for mononuclear vanadium(V) species[15] (Supplementary Tables 1,2), the [VO$_2$(IDO)]$^{2-}$ complex is the dominant species at [H$_3$IDO]/[V] = 1 concentration ratio over the pH range of 4–14. This is an unexpected result, since related complexes containing a fully (triply) deprotonated cyclic imide-dioxime ligand have not been observed for other metal ions, including U(VI), Fe(III), Cu(II), Pb(II), and Ni(II)[18,19]. Figure 2a shows that a fully deprotonated cyclic glutaroimide-dioxime ligand (IDO$^{3-}$) coordinates to the V center through a tridentate-binding motif via the imide N atom and the oxime O atoms. The tridentate coordination is preserved under subsequent protonation of [VO$_2$(IDO)]$^{2-}$, as exemplified by the DFT optimized geometries of the [VO(OH)(IDO)]$^-$ and [VO(OH)(HIDO)] complexes (Fig. 2a). It should be noted that the [VO(OH)(IDO)]$^-$ tautomer with a proton located on a vanadium oxido moiety was found to be only 1.5 kcal/mol more stable than [VO$_2$(HIDO)]$^-$, where a proton resides on an oxime nitrogen of the ligand. This is consistent with previously reported crystallographic and spectroscopic studies of the 1:1 V(V)/HIDO$^{2-}$ complex, supporting the proposed existence of two tautomeric forms in aqueous solution[20].

Interactions of V(V) with two IDO$^{3-}$ ligands lead to the formation of a rare non-oxido [V(IDO)$_2$]$^-$ complex (Fig. 2a), which, in agreement with the single-crystal X-ray diffraction (XRD) data[20], consists of a bare V$^{5+}$ center bound to two fully deprotonated glutaroimide-dioximes in a highly distorted octahedral environment. The distortion of [V(IDO)$_2$]$^-$ from a centrosymmetric configuration is a noteworthy structural feature potentially affording insight to the origin of the strength and selectivity of V binding by the imide-dioxime ligand. To obtain more insight, ab initio molecular dynamics (AIMD) simulations were performed on the [V(IDO)$_2$]$^-$ complex inside a periodic box containing 84 water solvent molecules and one Na$^+$ counterion. The system was equilibrated for 5 ps at 300 K, followed by a 5 ps production run (see Methods). The simulations confirmed the stability of [V(IDO)$_2$]$^-$, with the AIMD trajectory showing a similar V(V) coordination environment with average V–N and V–O bond distances of 1.98 Å and 1.90 Å, respectively, in excellent agreement with the XRD determined average V–N (1.96 Å) and V–O (1.89 Å) bond lengths[20]. These results eliminate the possibility that crystal packing forces in the [V(IDO)$_2$]$^-$ crystal structure[20] are responsible for the deviation from an octahedral geometry, suggesting that the observed structural distortion originates from electronic effects. Molecular orbital calculations and application of vibronic coupling theory[21] to the [V(IDO)$_2$]$^-$ cluster revealed that the distortion is due to the pseudo Jahn-Teller effect (Supplementary Fig. 2 and Supplementary Note 2), the appearance of which is driven by the energy gain from increased metal-ligand covalent bonding[21]. Thus, the [V(IDO)$_2$]$^-$ complex naturally adopts a highly distorted geometry that enables stronger covalent interactions between IDO$^{3-}$ and V(V).

To provide further evidence for the formation of the highly stable non-oxido [V(IDO)$_2$]$^-$ complex in aqueous solution, we performed high-level CCSD(T)/aug-cc-pVDZ//M06/SSC/6-311 + + G** calculations (Supplementary Table 3) to determine the change in free energy, $\Delta G_{\text{aq}}$, for the equilibrium shown by Eq. 1:

$$[\text{VO(OH)(HIDO)}]^- + [\text{VO(OH)(HIDO)}]^- \Leftrightarrow [\text{V(IDO)}_2]^- + [\text{H}_2\text{VO}_4]^-$$

(1)

The calculations indicate that the formation of the 1:2 V(V)/IDO$^{3-}$ complex in the above reaction is a thermodynamically favorable process ($\Delta G_{\text{aq}} = -3.08$ kcal/mol). This is in contrast to

**Table 1 Theoretically calculated and experimental stability constants (log $\beta$) of vanadium complexes with glutaroimide-dioxime (H$_3$IDO) and acetamidoxime (HAO) ligands**

| Aqueous Species, Reaction | log $\beta^{theor\ a}$ | log $\beta^{expt\ b}$ |
|---|---|---|
| *Glutaroimide-dioxime (H$_3$IDO) ligand* | | |
| $H_2VO_4^- + H^+ + HIDO^{2-} \rightleftharpoons [VO_2(IDO)]^{2-} + 2H_2O$ | 22.9 | $21.2 \pm 0.4$ |
| $H_2VO_4^- + 2H^+ + HIDO^{2-} \rightleftharpoons [VO(OH)(IDO)]^- + 2H_2O$ | 25.6 | n.a |
| $H_2VO_4^- + 3H^+ + HIDO^{2-} \rightleftharpoons [VO(OH)(HIDO)] + 2H_2O$ | 29.0 | n.a. |
| $H_2VO_4^- + 2H^+ + 2HIDO^{2-} \rightleftharpoons [VO(IDO)_2]^{3-} + 3H_2O$ | n.a. | $35.9 \pm 0.8$ |
| $H_2VO_4^- + 4H^+ + 2HIDO^{2-} \rightleftharpoons [V(IDO)_2]^- + 4H_2O$ | 53.5 | $53.0 \pm 0.4$ |
| $H_2VO_4^- + 5H^+ + 2HIDO^{2-} \rightleftharpoons [V(IDO)(HIDO)] + 4H_2O$ | 54.9 | n.a. |
| $H_2VO_4^- + 6H^+ + 2HIDO^{2-} \rightleftharpoons [V(HIDO)_2]^+ + 4H_2O$ | 51.7 | n.a. |
| *Acetamidoxime (HAO) ligand* | | |
| $H_2VO_4^- + 2H^+ + AO^- \rightleftharpoons [VO_2(AO)] + 2H_2O$ | 17.7 | n.a. |
| $H_2VO_4^- + 2H^+ + 2AO^- \rightleftharpoons [VO_2(AO)_2]^- + 2H_2O$ | 25.5 | n.a. |
| $H_2VO_4^- + 3H^+ + 2AO^- \rightleftharpoons [VO(OH)(AO)_2] + 2H_2O$ | 28.5 | n.a. |
| $H_2VO_4^- + 4H^+ + 3AO^- \rightleftharpoons [V(AO)(H_{-1}AO)_2] + 4H_2O$ | 43.4 | n.a. |
| $H_2VO_4^- + 6H^+ + 3AO^- \rightleftharpoons [V(AO)_3]^{2+} + 4H_2O$ | 39.5 | n.a. |

$^a$Obtained at 25 °C and $I = 0$
$^b$Obtained at 25 °C and $I = 0.5$ M

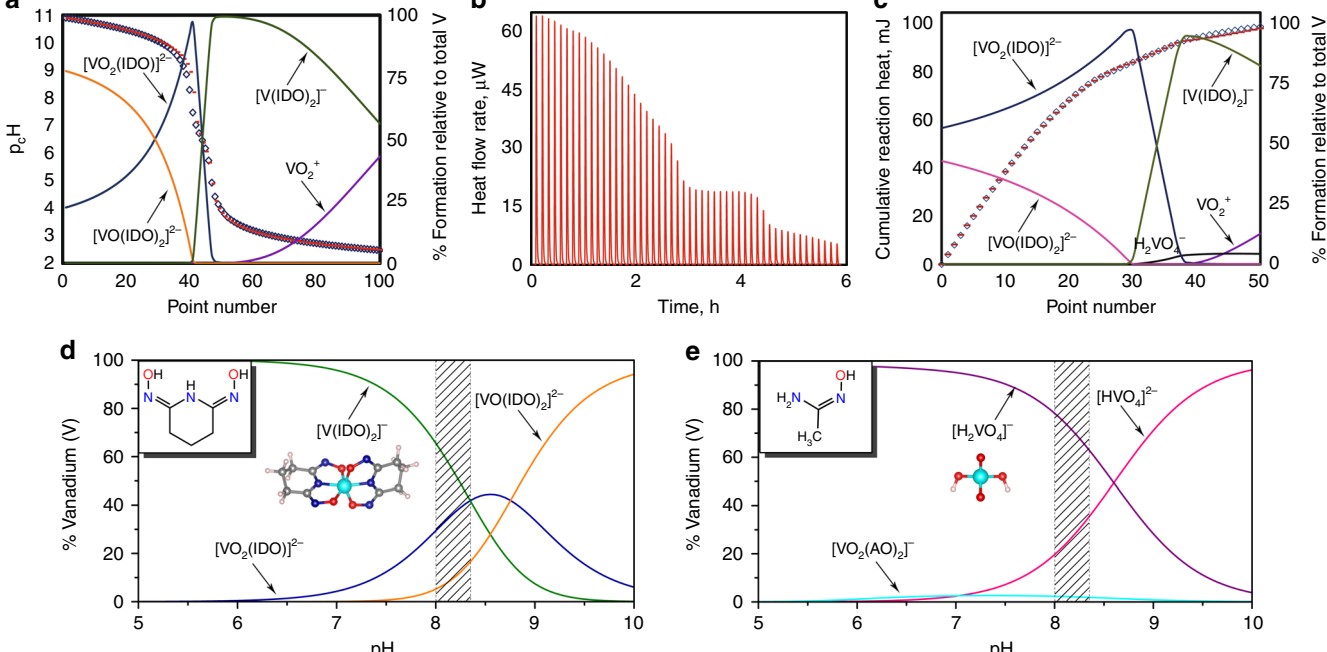

**Fig. 3** Potentiometric and calorimetric titrations along with V(V) speciation diagrams under seawater conditions. **a** Potentiometric titration for the complexation of vanadium with H$_3$IDO at 25 °C and $I = 0.5$ M (NaCl); observed (diamonds) and calculated (−) pH (left axis) with corresponding speciation (right axis); titration conditions: $V_o = 20.2$ mL, [V] = 1.01 mM, [H$_3$IDO] = 10.3 mM, [H] = 0.0129 M, titrant: 0.100 M HCl. **b** Calorimetric titration thermogram showing heat flow rate after each injection. **c** Cumulative observed (diamonds) and calculated (−) heats (left axis) with corresponding speciation (right axis) per injection; titration conditions: $V_o = 0.750$ mL, [V] = 1.0 mM, [H$_3$IDO] = 4.16 mM, [H] = 3.32 mM; titrant: 0.020 M HCl; 5 µL per injection. **d** Speciation of vanadium under seawater conditions with H$_3$IDO and **e** HAO ligands; [V] = 3.6 × 10$^{-8}$ M, [H$_3$IDO] = 0.5 M, [HAO] = 0.5 M. Structures of prevalent species at the seawater pH are visualized for clarity

typical metal binding thermodynamics, in which the binding of each successive ligand becomes weaker. By combining CCSD(T), DFT, and experimental stability constants[15] for a set of reactions, as described in Supplementary Note 1, we obtained a log $\beta^{theor}$ value of 53.5 for [V(IDO)$_2$]$^-$ (Table 1). To the best of our knowledge, this is the highest known stability constant for any V(V) species. The closest analog of H$_3$IDO, the bis-(hydroxylamino)−1,3,5-triazine ligand, can only form a single 1:1 dioxido-vanadium(V) complex with a log $\beta$ of ~25.2[22], while the non-oxido 1:3 V(V) complex with the trishydroxamate ligand, deferoxamine B, achieves a log $\beta$ value of ~45.0[23].

Another functional group of the polyamidoxime adsorbent, open-chain amidoxime (HAO), prefers to bind V(V) in a chelating fashion (Fig. 2b), whereas corresponding 2:1 complexes reveal coordination to V(V) in either monodentate ([VO$_2$(AO)$_2$]$^-$) or bidentate ([VO(OH)(AO)$_2$]) modes, depending on the degree of protonation (the full list of the lowest isomers is shown in Supplementary Fig. 3). Although there is no experimental evidence for the displacement of the V = O oxido bonds by HAO, we considered the possible formation of a hypothetical 1:3 non-oxido [V(AO)$_3$]$^{2+}$ complex to provide a complete picture of the V(V) complexation by HAO ligands.

Furthermore, since one of the prominent features of glutaroimide-dioxime ($H_3IDO$) is the ability to be fully deprotonated in the $[V(IDO)_2]^-$ complex, it is important to determine if further deprotonation of $AO^-$ is possible. The $[V(AO)_3]^{2+}$ complex was found to be stable to deprotonation after a 10 ps AIMD run in a cubic box containing 62 water molecules and two chloride ($Cl^-$) counterions. Only in the presence of two hydroxide ($HO^-$) counterions, corresponding to a highly alkaline solution, did the oxime nitrogen from two of the chelating $AO^-$ ligands lose a proton during the first 2 ps of AIMD, affording the neutral $[V(AO)(H_{-1}AO)_2]$ complex depicted in Fig. 2b. In striking contrast to cyclic imide-dioxime ($H_3IDO$), the results for open-chain amidoxime (HAO) revealed that, despite a large excess of the ligand in speciation modeling (Supplementary Fig. 4), no complex formation was observed over the entire pH range, indicating that all identified V(V)/HAO complexes were completely suppressed by stable anionic V(V) hydrolysis species, such as $H_2VO_4^-$ and $HVO_4^{2-}$.

**Potentiometric and calorimetric titrations**. Thermodynamic investigations, including potentiometric and calorimetric titration techniques, were performed to verify the computationally predicted complexes and log $\beta$ values (see Methods). Figure 3a shows a representative potentiometric titration for the vanadate-glutaroimide-dioxime ($H_3IDO$) system, which is best fit with $[VO_2(IDO)]^{2-}$, $[VO_2(H_2IDO)_2]^-$ (or $[V(IDO)_2]^- \cdot 2H_2O$), and $[VO_2(HIDO)_2]^{3-}$ (or $[VO(IDO)_2]^{3-} \cdot H_2O$) species, and Table 1 lists their associated equilibrium constants, log $\beta^{expt}$. As can be seen, $[VO_2(IDO)]^{2-}$ is dominant in high pH solutions; this observation is consistent with multinuclear $^{51}V/^{17}O$ NMR and X-ray crystallography results[20], indicating the $[VO_2(IDO)]^{2-}$ complex exists in solutions at relatively high pH, but crystallizes as $Na[VO_2(HIDO)]$, a protonated form of $[VO_2(IDO)]^{2-}$.

At lower pH, the 1:2 V(V)/glutaroimide-dioxime complex, $[VO_2(H_2IDO)_2]^-$, becomes the dominant species (Fig. 2a). This complex is equivalent to the non-oxido 1:2 complex, $[V(IDO)_2]^- \cdot 2H_2O$, formed by successive protonation and loss of oxido moieties as water. Though potentiometry alone cannot distinguish between these two forms, our computational results and the experimental $Na[V(IDO)_2] \cdot 2H_2O$ crystal structure[20] support this assertion. ESI-MS and $^{51}V/^{17}O$ NMR spectra of aqueous solutions containing $NaVO_3$ and glutaroimide-dioxime confirmed that the $[V(IDO)_2]^-$ complex forms under nearly the same conditions[20] as the $[VO_2(H_2IDO)_2]^-$ complex identified in the present study. Furthermore, the proposed mechanism for formation of the non-oxido V(V) complex via loss of the oxido ligands as water is supported by reports of other non-oxido V(IV) and V(V) complexes formed via a similar mechanism[23,24].

Comparison of log $\beta^{expt}$ and log $\beta^{theor}$ values of the vanadium complexes with $H_3IDO$ demonstrates near perfect agreement (Table 1). The slightly larger deviation between the experimental and theoretical results for the doubly charged $[VO_2(IDO)]^{2-}$ species compared to $[V(IDO)_2]^-$ can be attributed to the difference in the ionic strengths ($I$) of the solutions employed in the computations ($I = 0$ M) and the experiment ($I = 0.5$ M). We found that the inclusion of additional $[VO_2(HIDO)_2]^{3-}$ (or $[VO(IDO)_2]^{3-} \cdot H_2O$) in the model improved the fit of the potentiometric data in the high-pH region (Fig. 3a). This complex was not identified by our computations and any attempts to obtain the corresponding crystals were unsuccessful. Structural identification of this complex is difficult because it is a minor component in solution with $pC_H < 10$ (Fig. 3a) and because the possible ingress of carbon dioxide in solutions at high

pH complicates experiments. From a computational point of view, applying our protocol to predict log $\beta$ for $[VO(IDO)_2]^{3-}$ is not expected to give accurate results because of the known deficiencies in the implicit solvation model when treating highly charged species[25]. Nevertheless, it is reasonable to suggest that this complex could contain a short V=O bond with one of the glutaroimide-dioxime ligands coordinating to V in a tridentate mode, while the other binds to V in a monodentate fashion (Supplementary Fig. 5). Figure 3b shows a representative thermogram along with its associated calorimetric titration data (Fig. 3c) fitted using the three complexes identified from potentiometric titration. The formation of each of the complexes is highly favorable as evidenced by the strongly exothermic enthalpies (Supplementary Table 4; Supplementary Note 3).

In direct contrast to the $H_3IDO$ system, potentiometric titrations for the vanadate-HAO system did not show any complex formation over a wide range of pH values. Additional spectroscopic results, including $^1H$ and $^{51}V$ NMR data (Supplementary Fig. 6; Supplementary Note 4) also did not reveal any interaction between the two species under a wide range of experimental conditions: (1) pH 7.5–10 with $[V]_{total} = 7.5$ mM, $[HAO] = 15$ mM ($[HAO]/[V]$ ratio = 2); and (2) pH 8.5 with $[V]_{total} = 7.5$ mM, $[HAO] = 45$ mM ($[HAO]/[V]$ ratio = 6). This negative result is fully consistent with the conclusion drawn from our ab initio calculations.

Under seawater conditions (pH $\sim$ 8.1–8.3), vanadium(V) species primarily exists as $H_2VO_4^-/HVO_4^{2-}$ anions[26]. Therefore, the functional group of polyamidoxime adsorbent responsible for the effective sequestration of vanadium must be able to compete with $H_2VO_4^-/HVO_4^{2-}$ for the complexation of V(V). The speciation of vanadium under seawater conditions ($[V] = 3.6 \times 10^{-8}$ M) was calculated with the modeling program HySS[27] from the stability constants of V(V) complexes with $H_3IDO$ and HAO obtained in this work (Table 1) and the V(V) hydrolysis species taken from the literature[15,28] (Fig. 3d, e). At seawater pH V(V) is fully complexed, even in the presence of very dilute 0.001 M $H_3IDO$. In contrast, HAO forms no complexes with V(V) due to being suppressed by $H_2VO_4^-/HVO_4^{2-}$. As shown in Fig. 3e, increasing the HAO concentration to 0.5 M made little difference in the species distribution diagram, forming only a small fraction of the $[VO_2(AO)_2]^-$ complex at between pH 6.5 and 8.5. However, in the presence of 0.5 M $H_3IDO$, 100% of V(V) is complexed, predominantly forming the 1:2 $[V(IDO)_2]^-$ species at pH 5.0–8.5 (Fig. 3d). This is a particularly important finding as it suggests that the cyclic imide-dioxime functionality (Fig. 1c) is exclusively responsible for the extremely strong and selective adsorption of vanadium by amidoxime-functionalized polymers.

**X-ray absorption fine structure spectroscopy**. The corroboration between the computational and potentiometric titration results affords great confidence in the metal-binding behavior of $H_3IDO$ and HAO small molecules. Nevertheless, any inference from such work of V binding by polyamidoxime adsorbents is inherently predicated upon the assumption that graft polymers behave analogously to their monomeric precursors. Polymer morphology effects attributable to graft chain density, polydispersity, and solvophilicity have all been documented[29], and seawater constitutes a highly complex and variable matrix replete with fluctuations in metal concentration, dissolved organic content, salinity, pH, temperature and biological activity. Accordingly, direct validation of the V-binding environment on representative seawater-contacted adsorbents is indispensable for drawing any definitive conclusions.

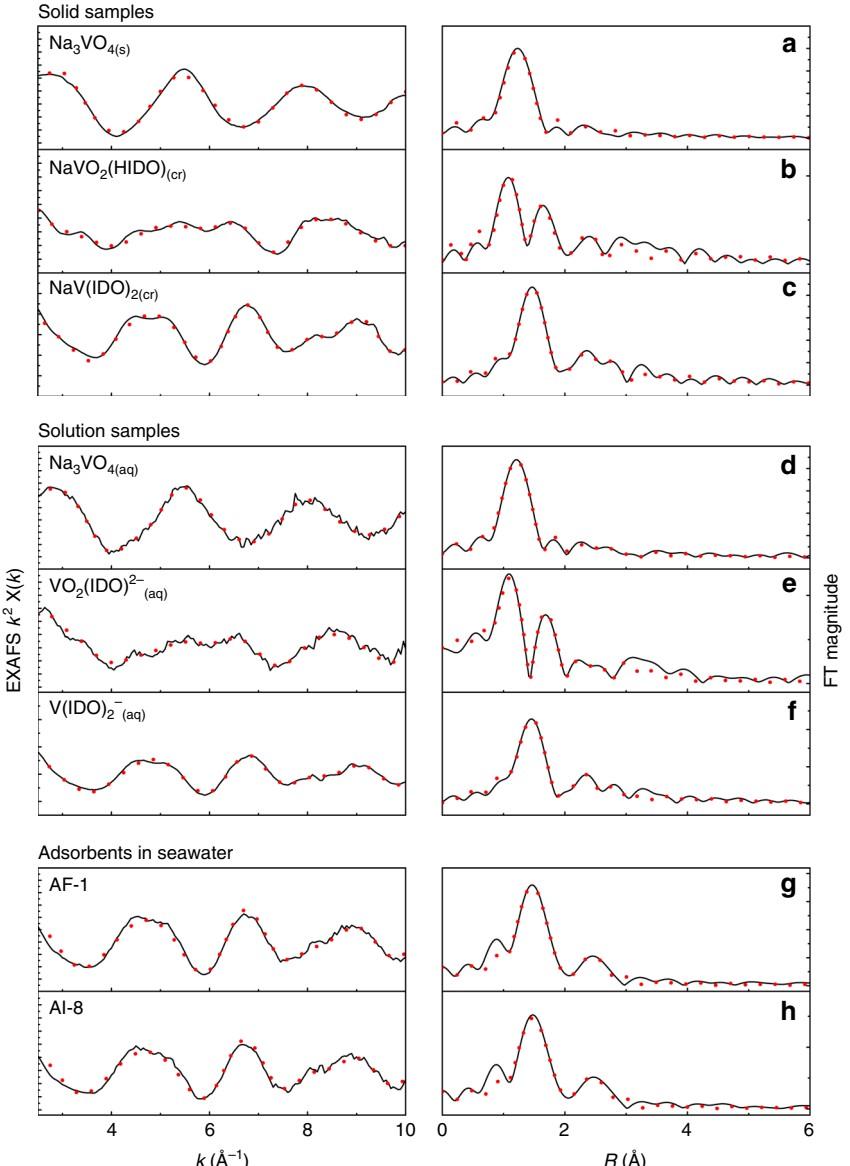

**Fig. 4** EXAFS spectra and magnitude of the Fourier transform of small molecule standards, solution samples, and seawater-contacted adsorbents. Solid samples: **a** $Na_3VO_{4(s)}$. **b** $Na[VO_2(HIDO)]_{(cr)}$. **c** $Na[V(IDO)_2]_{(cr)}$. Solution samples: **d** $Na_3VO_4$. **e** $[VO_2(IDO)]^{2-}$. **f** $[V(IDO)_2]^{-}$. Seawater-contacted adsorbents: **g** AF-1, **h** AI-8. All samples show the data as a solid black line and fit as a dotted red line

Although V is extracted at significantly greater concentrations than other transition metals, it nevertheless remains below the limits of detection for virtually all traditional spectroscopies. Furthermore, binding of numerous other metals and the absence of any long-range ordering impede x-ray/neutron scattering approaches. However, XAFS is devoid of such sample form limitations and simultaneously possesses sufficient sensitivity to provide the meaningful structural data from environmentally deployed adsorbent polymers.

Adsorbents used in XAFS analysis were prepared from poly (acrylamidoxime) co-polymerized with itaconic acid or vinyl phosphonic acid, colloquially referred to as AF1 and AI8, respectively[7,8]. Samples were conditioned with KOH and contacted for 56 days at the Marine Sciences Laboratory of Pacific Northwest National Laboratory, as reported in previous publications[7,8,14]. Metal uptake is reported in Supplementary Table 5. Polymer samples were prepared for XAFS analysis as reported previously[30,31] (see Methods), while small molecule standards of crystalline $Na[VO_2(HIDO)]$, $Na[V(IDO)_2]$ and

several vanadium oxides were blended with BN and pressed into self-supporting pellets. Samples of $Na_3VO_4$, and V(V) with $H_3IDO$ in a 0.1 M NaOH solution were also prepared, with the concentration of $H_3IDO$ selected to afford the $[VO_2(IDO)]^{2-}$ and $[V(IDO)_2]^{-}$ species, as per potentiometric titration data presented above.

Comparison of the X-ray absorption near edge (XANES) spectra for the seawater-contacted adsorbents, vanadium oxide standards, and $Na[V(IDO)_2]$ is particularly revealing (Supplementary Fig. 7). A common pre-edge feature at 5470 eV is apparent for both seawater-contacted adsorbents and $Na[V(IDO)_2]$. This feature is attributable to the $1s \rightarrow 3d$ transition arising from the distorted octahedral V coordination environment, identified computationally as due to the pseudo Jahn-Teller effect. The first derivative of the absorption spectrum reveals that the absorption edge, $E_0$, for these samples is located at 5482 eV. A similar $E_0$ is identified in the V(V) metal oxide standard, $Na_3VO_4$, particularly when compared to V(IV)

and V(III) species. Although anticipated from the literature[32] and previous computational work[33–35], this is the first instance the + 5 oxidation state is experimentally confirmed for adsorbent-bound V.

XANES analysis provides only preliminary support for the small molecule determined binding mode, whereas fits of the extended XAFS (EXAFS) region afford greater rigor and determination of a more precise V-binding environment (Fig. 4). Inspection of the $k^2\chi(k)$ data reveals similar oscillations between crystalline and aqueous $[VO_2(IDO)]^{2-}$, as well as between $[V(IDO)_2]^-$ samples and seawater-contacted adsorbents, suggesting common local V coordination environments. Fits of $Na_2[VO_2(IDO)]$ and $Na[V(IDO)_2]$ were achieved using scattering paths calculated from the corresponding crystal structures. The same models were also used to fit the aqueous solutions of the analogous complexes, corroborating the modest effects of crystal packing interactions on the V local coordination environment, again as predicted computationally. Nevertheless, for the seawater-contacted adsorbents the scattering paths from $Na[V(IDO)_2]$ required elongation by approximately 0.1 Å to obtain a clean superposition of the experimental features observed in the data; a fit of the data was then obtained readily, achieving good statistics and reasonable physical parameters. As this modest distortion was not observed between the crystal structure and the aqueous small molecule sample, it is proposed to be attributable to morphology effects associated with greater steric encumbrance arising from inclusion of the $H_3IDO$ species within the polymer. Fit parameters for all data sets are provided in Supplementary Tables 6–14.

To eliminate the possibility of other potential V-binding modes contributing significantly to the EXAFS spectra for the seawater-contacted adsorbents, structure models were generated from the computationally optimized structures displaying alternative binding motifs (Supplementary Fig. 8). Calculation of the theoretical scattering paths from the atoms in the first coordination sphere was performed using FEFF 9[36], and reveals that all potential binding modes, including a vanadium oxido or hydroxide species yield an intense scattering path at low radial distance and are inconsistent with the experimental data for seawater contacted adsorbents. The combination of a good-quality fit from the $Na[V(IDO)_2]$ structure model, in conjunction with the negative results obtained for alternate models definitively confirms the V-binding mode on the seawater-contacted adsorbents is consistent with the small molecule behavior identified by computational investigation and thermochemical titrations.

## Discussion

The combined computational and experimental findings provided here represent a significant step toward a fundamental understanding of the selective adsorption of metal ions from seawater by polymeric adsorbent materials. To underscore the importance of the obtained results, it is instructive to briefly summarize the previous work on adsorbent development for the recovery of metals from the oceans. The majority of studies in this area were mostly focused on the extraction of uranium from seawater, with considerable experimental work done in the 1990 s in Japan, where repeated screening programs identified amidoxime-functionalized polymers as the most promising candidate adsorbent[37]. While recently improved ligand grafting techniques[38], manipulations with the fiber morphology[11], and the use of a combination of an amidoximated carbon electrode with the half-wave rectified alternating current electrochemical (HW-ACE) method[39] helped increase the overall adsorption capacity, there is still very little fundamental understanding of

how the polyamidoxime adsorbents function. For instance, it was not clear, which factors determine the selectivity of amidoxime-based adsorbents toward a particular metal ion in seawater. Although many computational and experimental works[17,30,31,40–42] have been performed to elucidate mechanisms of binding and accumulation of uranium on the adsorbent, a number of marine studies have shown that vanadium consistently outcompetes uranium and many other cations for amidoxime sorption sites[9]. The vanadium adsorption capacity of ~15 g per kg adsorbent (Fig. 1b) is indeed remarkable considering the small atomic weight of vanadium compared to uranium, its low concentration (1.9 p.p.b. or $3.6 \times 10^{-8}$ M), and the competition with a pool of other metal ions in seawater.

To provide insight into the unusually strong and selective adsorption of vanadium, we first identified and characterized possible complexes that could be formed between V(V) and the representative functional groups of typical polyamidoxime sorbents in aqueous solution. Unexpectedly, our computational studies revealed a substantial difference between the binding strengths of cyclic imide-dioxime and open-chain amidoxime functionalities toward vanadium. Quantum chemical calculations and potentiometric titrations established the formation of a rare 1:2 non-oxido V(V)/imide-dioxime complex with the highest known stability constant for V(V) species. However, no binding with the open-chain amidoxime was observed under experimental conditions. These results were indeed surprising, because uranium, for example, can bind to both functionalities, producing the respective U(VI)/imide-dioxime[18] and U(VI)/amidoxime[40–42] complexes. To validate the theoretical and experimental studies in aqueous solution and confirm the extension of small molecule data to bulk materials, seawater-contacted amidoxime-functionalized adsorbents were analyzed by EXAFS, revealing a V(V) coordination environment composed of two imide-dioxime functionalities.

From these results, we conclude the cyclic imide-dioxime form is solely responsible for the extremely strong sorption of vanadium by polyamidoxime sorbents. Of equal importance, these results also partially rationalize the previously reported EXAFS data that reveal no cyclic imide-dioxime contribution to the uranium coordination environment[30,31]. The much higher stability of the non-oxido V(V) complex compared to the U(VI) complex with cyclic imide-dioxime (Supplementary Table 15) can explain why uranium cannot access the imide-dioxime sorption sites that are occupied by vanadium. Furthermore, the broad pH region where the V(V) complex is stable (Fig. 3d) also helps explain the difficulty of removing vanadium from amidoxime-based sorbents during the elution process, even under highly acidic conditions[43].

A remarkable agreement between the results of ab initio simulations, potentiometric titrations, and EXAFS studies demonstrates the ability of the integrated methodology to characterize and predict the behavior of complex polymeric sorbent systems. Beyond simply identifying the vanadium-binding coordination environment on polyamidoxime adsorbents, these results convey critical knowledge on the rational design of advanced materials with enhanced performance for the selective recovery of valuable metals from seawater. For instance, it is now evident that the selectivity of polyamidoxime adsorbents for U(VI) over V(V) could be significantly increased by minimizing the presence of the cyclic imide-dioxime moiety. Additionally, the negative reaction enthalpy for formation of the non-oxido V(V) complex under seawater conditions (Supplementary Table 16) implies less sorption of vanadium at higher temperatures, which is in contrast to the adsorption of uranium[18]. Thus, another important practical takeaway is that the opposing effects of temperature on the sorption of vanadium

and uranium could be exploited to achieve optimal uranium extraction.

A general conclusion that may be drawn from our studies is that, without considering other macroscopic factors applicable to polyamidoxime sorbents, the unusually strong binding of V(V) to cyclic imide-dioxime is responsible for the much higher sorption of vanadium relative to most other seawater cations, corroborating the decisive role of thermodynamics in the selective recovery of metals from seawater. Thus, it is obvious that the availability of a suitable extraction agent would, at least in principle, permit the recovery of any desired metal from the oceans. For example, there are marine creatures that already concentrate out specific metals including lead, mercury or vanadium[44]. As was mentioned earlier, some tunicates, depending on the species, can concentrate high levels of vanadium; however, the conclusive route for the accumulation of vanadium ions from seawater has not yet been revealed[44]. According to recent studies, vanabin proteins in tunicate blood cells strongly chelate only V(IV), rather than V(V), indicating that reduction takes place before the vanadium ion binds to the protein[45]. This is a seemingly bioenergetically unfavorable process and it has been suggested that reduction involves nicotinamide adenine dinucleotide phosphate (NADPH) as a reducing agent[45]. In contrast, cyclic imide-dioxime can easily capture V(V) from seawater in the form of the very stable non-oxido $[V(IDO)_2]^-$ complex. Taking cues from our studies, it is reasonable to assume that other yet-undiscovered vanadium-binding proteins that possess imide-dioxime-like coordination environments may exist in tunicates. Given that investigations into the fundamental coordination of V by different ligands are helping advance understanding of this complex biological process[45], we expect the results of our studies will also contribute toward elucidating the mechanisms of vanadium accumulation in the marine organisms.

## Methods

**Computational methods**. Quantum chemical calculations were performed with the Gaussian 09 D.01 software. The density functional theory (DFT) approach was adopted using the M06 density functional with the standard Stuttgart small-core (SSC) 1997 scalar relativistic effective core potential (RECP), the associated contracted (6 s/5p/3d/1 f) basis set for vanadium and the 6-311 + + G(d,p) basis set for all other elements. Vibrational frequency calculations were performed at the B3LYP/SSC/6-31 + G(d) level to ensure that geometries (optimized at the same level of theory) were minima and to compute zero-point energies and thermal corrections. Using the gas-phase geometries, implicit solvent corrections were obtained at 298 K with the SMD[46] solvation model, as implemented in Gaussian 09, at the B3LYP/SSC/6-31 + G(d) level of theory. The preference for using a combination of the M06 and the B3LYP functionals with the SMD solvation model was based on the results of our previous studies[17,47], which showed that the chosen levels of theory provide the best overall performance in predicting the log $\beta$ values of uranyl[47] and vanadium[17] ions complexes with anionic oxygen and amidoxime donor ligands. Single-point coupled-cluster theory calculations, CCSD(T)/aug-cc-pvDZ (the valence electrons on C, O, H and the valence and subvalence electrons (3 s, 3p) on V were correlated), using M06/SSC/6-311 + + G(d,p) optimized geometries, were employed for selected V complexes with $H_3IDO$ and HAO ligands. Complexation free energies in aqueous solution, $\Delta G_{aq}$, and stability constants, log $\beta^{theor}$, were calculated using the methodology, described in our previous work on V (V) and V(IV)-containing complexes[17]. The ab initio MD simulations in the Helmholtz ensemble (NVT) were performed using VASP (Vienna Ab initio Simulation Package), version 5.3.5. Additional details are provided as Supplementary Methods.

**Potentiometric and calorimetric titrations**. Potentiometric titrations were carried out using an automated titration system consisting of a thermostated glass vessel, Metrohm model 713 pH meter, and a Metrohm model 765 Dosimat. The temperature of the titration solutions in the glass vessel was maintained at 25.00 ± 0.01 °C using a circulating water bath. To minimize the ingress of $CO_2$, the solutions were blanketed with a gentle stream of argon. A Metrohm Unitrode combination glass electrode was used to measure the electrode potentials in the titrations.

A TAM III microcalorimeter (TA instruments) was used to measure the enthalpies of complexation for V(V) complexes with $H_3IDO$. The titration

assembly consists of two matching stainless steel vessels (1 mL total volume) as the reference and sample cells, a gold stirrer, and gas-tight syringes (0.250 mL total volume) equipped with either gold or stainless steel needles for titrant injections. During a given titration, the calorimeter measures the heat flow from the sample cell as the titrant is injected into the titration solution and compares it to that of the reference cell. Integrating the measured heat flow as a function of time for each injection interval generates the heat absorbed or released as a result of dilution and/or complexation. Additional details are provided as Supplementary Methods.

**XAFS sample preparation, measurements, and data analysis**. For seawater contacted fibers, ~100 mg wet fibers were washed with DI water, dried overnight at 40 °C in a vacuum oven, immersed in liquid nitrogen and pulverized with a mortar and pestle. Small molecule and vanadium oxide standards were loaded on an aluminum holder with rectangular opening of 20 mm (l) × 2 mm (w) and a thickness of 0.5–1.0 mm, sealed with Kapton tape. For liquid samples, a Teflon holder was used to contain the samples, sealed with Kapton tape. Polymer samples were enclosed in a Nylon washer of 4.953 mm inner diameter, pressed into a self-supporting pellet, and sealed on both sides with Kapton tape. This method of sample containment was approved by the APS Radiation Safety Review Board to ensure double containment for analysis of radioactive samples, as natural uranium extracted from seawater is mildly radioactive.

The XAFS data were collected at the V K-edge (5465 eV) on beamline 10-BM-B of the Advanced Photon Source for the polymers and vanadium oxide standards, and beamline 4-3 of the Stanford Synchrotron Radiation Lightsource for the liquid samples as well as $Na[VO_2(HIDO)]$ and $Na[V(IDO)_2]$ small-molecule standards. XAFS spectra of the vanadium oxides and small-molecule standards were collected in transmission mode, while the liquid samples and polymer samples were collected using a fluorescence detector.

The data were processed with the Athena and Artemis programs of the IFEFFIT package[48]. The reference foil data were aligned to the first zero-crossing of the second derivative of normalized $\mu(E)$ data, which was calibrated to the literature $E_0$ value for the vanadium K-edge. Spectra were averaged in $\mu(E)$ prior to normalization. Background removal was achieved by spline fitting. Additional details are provided as Supplementary Methods and Supplementary Table 17.

**Data availability**. All the data supporting the findings discussed here are available within the paper and its Supplementary Information files, or from the corresponding authors upon request.

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

## Acknowledgements

The computational work (A.S.I. and V.S.B.) used resources of the National Energy Research Scientific Computing Center, a DOE Office of Science User Facility supported by the Office of Science of the U.S. Department of Energy under Contract No. DE-AC02-05CH11231. XAFS work (C.W.A.) used the Advanced Photon Source, a U.S. Department of Energy (DOE) Office of Science User Facility operated for the DOE Office of Science by Argonne National Laboratory under Contract No. DE-AC02-06CH11357 and (Z.Z. and L.R.) the Stanford Synchrotron Radiation Lightsource, SLAC National Accelerator Laboratory, supported by the U.S. Department of Energy, Office of Science, Office of Basic Energy Sciences under Contract No. DE-AC02-76SF00515. The thermodynamic experimental work (potentiometric and calorimetric titrations, C.J. L. and L.R.), collection and analysis of the EXAFS data for seawater-contacted adsorbents (C.W.A.) and all computational studies (A.S.I. and V.S.B.) were supported by the Fuel Cycle Research and Development Campaign (FCRD)/Fuel Resources Program, Office of Nuclear Energy, U.S. Department of Energy (USDOE). Experimental work by B.F.P. and J.A. was supported by the Nuclear Energy University Program (NEUP) at the University of California, Berkeley (UCB). Collection and analysis of the EXAFS data for small-molecule standards (Z.Z. and L.R.) was supported by USDOE, Office of Science, Office of Basic Energy Sciences, the Heavy Element Chemistry Program at Lawrence Berkeley National Laboratory. The work at Oak Ridge National Laboratory was sponsored by the U.S. Department of Energy, Office of Nuclear Energy, under Contract DE-AC05-00OR22725 with Oak Ridge National Laboratory, managed by UT-Battelle, LLC. The work at Lawrence Berkeley National Laboratory was supported by the U.S. Department of Energy, Office of Nuclear Energy and Office of Science under Contract No. DE-AC02-05CH11231 with Lawrence Berkeley National Laboratory, managed by University of California.

## Author contributions

A.S.I.: Performed the computations. C.J.L.: Conducted the potentiometric and calorimetric titrations of V(V)/glutaroimide dioxime systems. C.W.A.: Designed, performed and analyzed the XAFS experiments for seawater-exposed fibers. Z.Z. and L.R.: Conducted the EXAFS experiments for the solid and solution samples of orthovanadate,

the 1:1 and 1:2 V(V)/glutaroimide dioxime complexes and Z.Z. analyzed the EXAFS data. L.R.: Supervised the work by C.J.L. and Z.Z at LBNL. B.F.P.: Designed and performed the NMR experiments and titrations with acetamidoxime (HAO), synthesized Na [VO$_2$(HIDO)]$_{(cr)}$ and Na[V(IDO)$_2$]$_{(cr)}$ for EXAFS studies, and assisted Z.Z. in EXAFS sample preparation (performed at SSRL). J.A.: Supervised the research of B.F.P. at University of California, Berkeley, and at LBNL. The manuscript was written through contributions from all authors. V.S.B. and L.R.: Conceived and designed the computations and experiments, respectively. All authors discussed the results and commented on the manuscript.

## Additional information

**Competing interests:** The authors declare no competing financial interests.

