## [Peer Review File · Nature Communications]

REVIEWERS' COMMENTS:

Reviewer #1 (Remarks to the Author):

This manuscript reports on a combined experimental/theoretical study of the selective binding of Vanadium by polyamidoximes posing an impediment for the U extraction from seawaters upon using these materials. This work is of potential interest and it is very well executed and written. The methods and techniques used are adequate enough to serve the present purpose. This study could be used to improve the performance of polyamidoximes in capturing uranyl from seawaters as well to form a theoretical basis to further elucidated the V capture from marine organisms. I do recommend publication in Nature Communications as is.

Reviewer #2 (Remarks to the Author):

The present work describes a detailed study focused on the unusually strong and selective binding of V by polyamidoximes. The combination of theory and experiment aimed at understanding the high stability constants and unusual formation of "naked" V+5 complexes as well as the affinity of V-species for cyclic amidoximes. The preparation methods and characterization procedures are sound and clear. The present manuscript is well written and referenced while the reported findings are very interesting and are expected to attract the interest of synthetic and material chemists, marine biologists and theoreticians. The findings here are also useful for our understanding of vanadium's biochemistry not only for marine organisms but also in the case of Amavadin which is present in mushroom species.

Thus, I am happy to recommend publication of the present manuscript in Nature Comm.

Reviewer #3 (Remarks to the Author):

This paper is long on theory and claims and short on results.

The authors come to the unremarkable conclusion that "...the unusually strong binding of V(V) to cyclic imide-dioxime is responsible for the much higher sorption of vanadium relative to most other seawater cations..." and that thermodynamics may be important in extraction.

This relatively modest advance leads the authors to conclude that "The combined computational and experimental findings provided here represent a significant step toward a fundamental understanding of the selective adsorption of metal ions from seawater by polymeric adsorbent materials."

In the introduction (page 3) it is claimed that this system "... achieves the highest stability constant values ever reported for V(V), exemplified by the formation of a rare non-oxido V5+ complex." Catechol complexes of vanadium(V) have been known for decades and there are many papers describing them.

Of course the vanadium tunicate complexes contain catechol chelating groups. The authors seem to imply that all tunicates utilize vanadium. This is not true, some use iron. The discussion on page 5 "...suggesting that the observed structural distortion originates from electronic effects..." is very unremarkable. This distortion has been seen many times before in pseudo octahedral complexes of V(IV) and V(V) complexes and the origins explained.

Reviewer #1:

General Comments:

This manuscript reports on a combined experimental/theoretical study of the selective binding of Vanadium by polyamidoximes posing an impediment for the U extraction from seawaters upon using these materials. This work is of potential interest and it is very well executed and written. The methods and techniques used are adequate enough to serve the present purpose. This study could be used to improve the performance of polyamidoximes in capturing uranyl from seawaters as well to form a theoretical basis to further elucidated the V capture from marine organisms. I do recommend publication in Nature Communications as is.

Response to General Comments: We are thankful to the reviewer for her/his favorable comments. The reviewer recommended to publish the paper as is.

Reviewer #2:

General Comments:

The present work describes a detailed study focused on the unusually strong and selective binding of V by polyamidoximes. The combination of theory and experiment aimed at understanding the high stability constants and unusual formation of “naked” V+5 complexes as well as the affinity of V-species for cyclic amidoximes. The preparation methods and characterization procedures are sound and clear. The present manuscript is well written and referenced while the reported findings are very interesting and are expected to attract the interest of synthetic and material chemists, marine biologists and theoreticians. The findings here are also useful for our understanding of vanadium’s biochemistry not only for marine organisms but also in the case of Amavadin which is present in mushroom species.

Thus, I am happy to recommend publication of the present manuscript in Nature Comm.

Response to General Comments: We are thankful to the reviewer for her/his favorable comments. The reviewer recommended to publish the paper as is.

Reviewer #3:

General Comments:

This paper is long on theory and claims and short on results. The authors come to the unremarkable conclusion that "...the unusually strong binding of V(V) to cyclic imide-dioxime is responsible for the much higher sorption of vanadium relative to most other seawater cations.." and that thermodynamics may be important in extraction. This relatively modest advance leads the authors to conclude that "The combined computational and experimental findings provided here represent a significant step toward a fundamental understanding of the selective adsorption of metal ions from seawater by polymeric adsorbent materials."

Response to General Comments: We appreciate the reviewer's comments on our paper. We believe our findings will be useful for a fundamental understanding and rational design of selective polymeric adsorbent materials. Our results indeed challenge the long-established methods of amidoxime-functionalized adsorbent preparation, enabling researches in the field to reconsider the existing routes for the adsorbent design and urge them to explore new chemical formulations that avoid the cyclic imide-dioxime moiety without compromising strong U-binding capability. In addition to the technological implications, our study will be of interest to the broad research community, including synthetic and bioinorganic chemists, because it provides an important example of the role of individual ligand functionalities in the metal ion recognition in complex aqueous solutions. In order to reiterate the importance of our results we have added the following sentences to the text [page 14 (bottom)]:

A remarkable agreement between the results of ab initio simulations, potentiometric titrations, and EXAFS studies demonstrates the ability of the integrated methodology to characterize and predict the behavior of complex polymeric sorbent systems. Beyond simply identifying the vanadium binding coordination environment on polyamidoxime adsorbents, these results convey critical knowledge on the rational design of advanced materials with enhanced performance for the selective recovery of valuable metals from seawater.

Specific Comments:

In the introduction (page 3) it is claimed that this system "... achieves the highest stability constant values ever reported for V(V), exemplified by the formation of a rare non-oxido V⁵⁺ complex." Catechol complexes of vanadium(V) have been known for decades and there are many papers describing them.

Response:

Unfortunately, our search of Critical Stability Constants by Martell and Smith and more recent literature did not show any reported aqueous formation constants for the V(V) complexes with catechol ligands. Part of the problem is that catechols are typically oxidized by V(V) in aqueous solution and the resulting complexes contain vanadium in either oxidation states III or IV. [For references please see: Crans, D. C. et al. The chemistry and biochemistry of vanadium and the biological activities exerted by vanadium compounds. *Chem.*

Rev. **104**, 849-902, (2004); Ryan, D. E. et al. Reactivity of tunichromes: reduction of vanadium (V) and vanadium (IV) to vanadium (III) at neutral pH. *J. Am. Chem. Soc.* **114**, 9659-9660, (1992); Kime-Hunt, E. et al. Vanadium metabolism in tunicates: The coordination chemistry of V (III), V (IV), and V (V) with models for the tunichromes. *J. Inorg. Biochem.* **41**, 125-141, (1991); Ryan, D. E. et al. Reactions between vanadium ions and biogenic reductants of tunicates: Spectroscopic probing for complexation and redox products in vitro. *Biochem.* **35**, 8651-8661, (1996); Branca, M. et al. Formation and structure of the tris (catecholato) vanadate (IV) complex in aqueous solution. *Inorg. Chem.* **29**, 1586-1589, (1990)]. One exception is a complex of bare V(V) with three di-tert-butylcatecholate ligands, $V(DBCat)_3^-$, but this was crystallized from nonaqueous solutions and its stability in aqueous solution is unknown. [Cass, M. E., Gordon, N. R., & Pierpont, C. G. Catecholate and semiquinone complexes of vanadium. Factors that direct charge distribution in metal-quinone complexes. *Inorg. Chem.* **25**, 3962-3967, (1986)]. In contrast, our study shows that the imide-dioxime non-oxido V(V) complex forms and persists in aqueous and complex seawater solutions, possessing the highest reported stability constant for V(V) in water to date.

Of course the vanadium tunicate complexes contain catechol chelating groups.

Response:

The reviewer's statement seems to rely on the data published some years ago claiming that the vanadium tunicate complexes contain catechol chelating groups. However, at the time of these early model studies it was not known that vanadium and tunichrome (containing catechol groups) were located in different cells, and that little vanadium would be complexed to tunichrome (please refer to the review article by Crans et al., [Crans, D. C., Smee, J. J., Gaidamauskas, E. & Yang, L., The chemistry and biochemistry of vanadium and the biological activities exerted by vanadium compounds. *Chem. Rev.* **104**, 849-902, (2004)] [ref. 45 in the main text of our article]).

The authors seem to imply that all tunicates utilize vanadium. This is not true, some use iron.

Response: We agree with the reviewer that not all tunicates (but still most of them) accumulate vanadium. We are thankful to the reviewer for this notice. We have modified the text accordingly [page 15 (bottom)]:

As was mentioned earlier, some tunicates, depending on the species, can concentrate high levels of vanadium, however, the conclusive route for the accumulation of vanadium ions from seawater has not yet been revealed.

The discussion on page 5 "...suggesting that the observed structural distortion originates from electronic effects.." is very unremarkable. This distortion has been seen many times before in pseudo octahedral complexes of V(IV) and V(V) complexes and the origins explained.

Response: The observed structural distortions in the non-oxido $[\text{V}(\text{IDO})_2]^-$ complex that we have attributed to the pseudo Jahn-Teller effect do not constitute the main results of this study. Nevertheless, we find it interesting and worth further including in the SI section that the observed distortions in the studied complex originate from vibronic coupling of ground and excited states that lower the energy of the distorted compared to the symmetric configuration. Identifying the origin of the distortion in $[\text{V}(\text{IDO})_2]^-$ could help to better understand why the imide-dioxime ligand is very effective in binding vanadium (V) ions in solution.